# Children Involved in Team Sports Show Superior Executive Function Compared to Their Peers Involved in Self-Paced Sports

**DOI:** 10.3390/children8040264

**Published:** 2021-03-30

**Authors:** Silke De Waelle, Felien Laureys, Matthieu Lenoir, Simon J. Bennett, Frederik J.A. Deconinck

**Affiliations:** 1Department of Movement and Sports Sciences, Ghent University, 9000 Ghent, Belgium; Felien.Laureys@UGent.be (F.L.); Matthieu.Lenoir@UGent.be (M.L.); Frederik.Deconinck@UGent.be (F.J.A.D.); 2Research Institute for Sport and Exercise Sciences, Liverpool John Moores University, Liverpool L3 3AF, UK; S.J.Bennett@ljmu.ac.uk

**Keywords:** executive function, athletes, development, children

## Abstract

Children’s motor and cognitive functions develop rapidly during childhood. Physical activity and executive function are intricately linked during this important developmental period, with physical activity interventions consistently proving to benefit children’s executive function. However, it is less clear which type of physical activity shows the strongest associations with executive function in children. Therefore, this study compared executive function performance of children aged 8 to 12 that either participated in team sports or self-paced sports or were not involved in any kind of organized sports (non-athletes). Results demonstrate that children participating in team sports show superior executive function compared to children participating in self-paced sports and non-athletes. Importantly, children participating in self-paced sports do not outperform non-athletes when it comes to executive function. This study is the first to show that even at a very young age, team sports athletes outperform athletes from self-paced sports as well as non-athletes on a multifaceted and comprehensive test battery for executive function. Furthermore, our findings support the hypothesis that cognitively engaging physical activity, such as participation in team sports, might show stronger associations with executive functioning compared to other types of sports and physical activity.

## 1. Introduction

Childhood is a critical period for children’s motor and cognitive development. Although they have been regarded as separate functions for a long time, there is now compelling evidence for an intricate relationship between both [1,2,3]. In this respect, it has been shown that motor control and cognitive function engage overlapping brain regions, e.g., parts of the prefrontal cortex and the cerebellum [2,4,5]. The prefrontal cortex is traditionally considered a crucial region for cognitive processing, whereas the cerebellum is heavily involved in motor control. The joint activation, therefore, supports the relationship between both functions. Moreover, an increasing number of studies have shown that motor training or physical activity interventions positively affect executive function [4,6], which represents a part of cognition and is defined as the control mechanism that is mainly involved in goal-directed behavior [7,8]. The present study builds upon this evidence and explores executive function in a sample of young female athletes from different sports as well as non-athletes.

Executive function is often categorized into three interrelated subcomponents: shifting, inhibition, and working memory [7]. Shifting concerns the ability to efficiently switch between different tasks; inhibition refers to the ability to inhibit preprogrammed responses; and working memory can be described as the ability to keep and manipulate task-relevant information in the short-term memory. The positive effect of exercise on executive function in children has been shown for both single bouts of exercise and longer exercise interventions (for a review, see [9]). For example, 20 min of aerobic exercise (treadmill walking) has been demonstrated to acutely improve children’s inhibition performance [10]. Furthermore, three months of daily aerobic exercise has also been shown to benefit children’s inhibition skills [11]. With regard to the neurophysiological mechanisms behind these effects, there is agreement that physical activity leads to elevated levels of growth factors, including brain-derived neurotrophic factor, which positively influences brain plasticity (neurogenesis and synaptic plasticity) [12]. This increased brain plasticity is observed in the hippocampus, a hub for memory-related processes, including executive function [9], and might be further enhanced due to the cognitive demands inherent to any kind of physical activity [12]. In addition, there is evidence that aerobic exercise alone is not the most efficient medium to improve executive function and that an extra cognitive component needs to be added to exercise for a maximized effect [13]. Therefore, it follows that learning complex coordinative movement patterns within the dynamic context of sports, and especially team sports, might be of particular value.

An interesting approach to explore this issue further is to consider the effect of participation in organized sports on executive function. For instance, participating in these sports will challenge children cognitively by requiring them to learn new and complex movement patterns. In this respect, a few studies have investigated the possible link between sports participation and executive function in children. In a longitudinal study in preschoolers (3 to 5 years old), McNeill and colleagues [14] found that children who participated in some form of organized sports did not show superior executive function one year later compared to those not involved in organized sports. However, since the period for rapid development of executive function occurs after preschool (i.e., from the age of six years old onward [15]), it is possible that the children in this study were too young to already show these associations between sports participation and executive function. Ishihara and colleagues [16], on the other hand, showed that 6-to-11-year-old children who participated in tennis lessons for one year improved their executive functions over that period. Furthermore, Formenti and colleagues [17] found that children who practiced an open-skill sport (e.g., soccer or volleyball) demonstrated superior inhibitory control compared to children practicing closed-skill sports (e.g., gymnastics or swimming) and sedentary children.

While the evidence for the link between sports participation and executive function in children is rather limited, this link has been established more clearly in adults. In this regard, athletes have consistently shown superior executive function compared to non-athletes (for a meta-analysis, see [18,19]). Moreover, team sports athletes (e.g., volleyball, soccer, hockey, etc.) seem to have an advantage in executive function compared to athletes from other sports. For example, a large-sampled study by Applebaum and colleagues [20] indicated that team sports athletes not only outperform non-athletes but also athletes from other sports on working memory tasks. Furthermore, Jacobsen and colleagues [21] also demonstrated that team sports players scored highest in problem-solving. However, in their study, athletes from self-paced sports (i.e., sports that allow the athlete time to prepare themselves for critical actions and perform at their own pace [22]) also showed superior inhibition performance. Thus, it seems that attunement to differing demands of specific sports types relates to superior performance on varying cognitive measures.

Thus, there seems to be a clear link between participation in different types of sports and executive function in adulthood, while there is considerably less evidence for this link in childhood. The study of Formenti and colleagues [17] has been the first to investigate this in a sample of 8-to-12-year-old children. However, their measurement of executive function consisted of only one inhibition task, whereas executive function is typically defined as a broad construct, containing at least three interrelated subcomponents (i.e., shifting, working memory, and inhibition) that are all measured by different tasks [7]. In children (under 12 years), although measured by a range of different tasks, executive function can best be defined as a unitary construct with a single factor that represents the multiple subcomponents of executive function [23,24]. Consequently, more than one test is needed to capture the construct of executive function in a comprehensive manner, regardless of the factor structure that is applied.

Therefore, the current study aims to further clarify the differences in executive function between children participating in different types of sports. Using seven different computer-based, neuropsychological tasks, we compare executive function in three groups of 8-to-12-year-old girls: athletes who are engaged in team sports; athletes from self-paced sports; and representative peers who are not involved in sports. Based on evidence that sports and physical activity have a positive effect on executive function, we expect that the groups involved in sports will demonstrate higher levels of executive function than their non-athletic peers. Extending upon the work by Formenti and colleagues [17], we also hypothesize that even at a young age, team sports athletes will show superior executive function performance than athletes from other sports and non-athletes. Such difference could indeed indicate that the context of team sports entails a higher level of cognitive engagement compared to self-paced sports.

## 2. Materials and Methods

A total of 170 girls between 8 and 12 years old were recruited for this comparative descriptive study. Participants were recruited at six Flemish elementary schools of various backgrounds (state schools, method schools, and catholic schools), thereby creating a convenient and representative sample of Flemish children. Participants were categorized into three different sports participation groups: (1) non-athletes: girls who did not participate in sports other than the physical education lessons at school, (2) self-paced sports: girls who participated in self-paced sports (cycling, swimming, or athletics) for at least 2 hours per week, and (3) team sports: girls who played team sports (basketball, volleyball, soccer, korfball, or hockey) for at least 2 hours per week. Table 1 displays the number of participants and the average age for each group.

Prior to the study, participants and their parents provided written informed consent and were made aware of the fact that they could withdraw from the study at any time without consequence. This research was reviewed by an independent ethical review board and conforms with the principles and applicable guidelines for the protection of human subjects in biomedical research.

To measure executive functioning, seven tests from the Cambridge Brain Sciences (CBS) test battery were selected. These tests are all based on well-validated neuropsychological tasks that have been adapted to be suitable for computerized testing [25]. The test battery has been used in several large-sampled studies, and its dynamically varying difficulty levels (i.e., difficulty of a trial decreases or increases depending on whether or not the previous response was correct) and adequate test-retest reliability (r = 0.68) makes it suitable for almost all ages and less sensitive to floor and ceiling effects [26,27]. To assess working memory, the Spatial Span, Monkey Ladder, and Token Search tests were used. For each test, the maximum recall was used as an outcome variable. To assess inhibition, the Double Trouble and Sustained Attention to Response tasks were used, where the percentage of correct responses was used as an outcome variable. To assess shifting performance, the Odd One Out task was used with the number of correct attempts as an outcome variable. Lastly, to assess planning, the Spatial Planning task was used with the total score as the outcome variable. A full description of the tasks and how their outcome measures are calculated can be found in Appendix A. The executive function test battery lasted about 20 min for each participant and was administered on a 9.7 inch Apple iPad 2017 that had to be held in an upright position. Before the test, participants received a general explanation of the test battery as well as detailed explanations before each test. A trained researcher was present to ensure the test was executed correctly and to answer any additional questions.

Since several studies have indicated that executive function is best described as a unitary construct in childhood [23,28], the current study has used a weighted sum score approach toward executive function. Before analysis, this weighted sum score for the executive function was calculated by individually weighting each of the seven tests based on the loadings from the benchmark model for executive functioning by Laureys and colleagues [24]. This benchmark model, validated on more than 2000 children and adolescents, employing the same tests that are used in the current study, indicates that between 8 and 12 years old, executive function can best be described as a unitary construct. Therefore, the current study also uses one weighted sum score to examine executive function in this age range. Detailed information about the model and the specific loadings can be found in Appendix B.

Differences in executive function between the different groups were analyzed using a one-way Analysis of Covariance (ANCOVA), with the group as the fixed factor and age as the covariate. The weighted sum score of executive function was used as the dependent variable representing executive function. Assumptions of normality and independence were checked before the analyses [29]. Furthermore, the Levene’s test was used to check the assumption of homogeneity of variances [30]. Lastly, the assumption of homogeneity of regression slopes was also checked [29]. Estimated marginal means were compared using the Bonferroni method. Effect sizes (partial eta square) were reported, and the significance level was set to *p* < 0.05.

## 3. Results

### 3.1. Descriptive Statistics

Table 2 provides an overview of the mean score for each test, as well as the mean weighted sum score for executive function, across the three groups. For more detail regarding the choice of outcome measures, their units, and their calculations, readers are referred to Appendix A. Visual inspection of the histograms as well as Shapiro–Wilk’s tests confirmed that the executive function sum score variable was normally distributed in the full sample (W(170) = 0.991, *p* = 0.366) as well as within each sports group (W_controls_(59) = 0.980, *p* = 0.435; W_self-paced sports_(25) = 0.941, *p* = 0.158; W_team sports_(86) = 0.991, *p* = 0.818) [29].

### 3.2. ANCOVA

The results of the Levene’s test confirmed the homogeneity of variances (F_2,167_ = 0.063, *p* = 0.939), and a one-way ANOVA confirmed that the covariate and the grouping variable were independent, as there was no difference in age between the different groups (F_2,169_ = 0.577, *p* = 0.574) [29]. The one-way ANCOVA demonstrated a significant effect of the covariate, indicating that there is a significant effect of age on executive function (F_1,166_ = 36.511, *p* < 0.001, *ηp*^2^ = 0.1803). Inspection of the interaction effect between age and sports group confirmed that the assumption of homogeneity of regression slopes was not violated, as there was no significant interaction (F_2,164_ = 0.551, *p* = 0.557). This indicates that the effect of the covariate was the same for all groups [29]. After controlling for the covariate, the main effect for group was also significant (F_2,166_ = 5.143, *p* = 0.007, *ηp*^2^ = 0.0584). The partial eta square effect size just fails to reach Cohen’s criteria for moderate effect sizes (0.588); however, considering the very small difference between Cohen’s cut-off criterion and our effect size (0.004), we consider this effect size moderate [31,32]. This indicates that, when controlling for the effect of age, the different sports groups differ in their executive function performance, with a moderate effect size. Bonferroni pairwise comparisons of the estimated marginal means revealed that team sports athletes significantly outperform non-athletes and athletes from self-paced sports (see Figure 1).

## 4. Discussion

The aim of this study was to compare performance on general executive function (i.e., treated as a unitary factor construct) of 8-to-12-year-old team sports athletes, athletes from self-paced sports, and non-athletes. The results of the current study show that team sports athletes demonstrate superior executive function performance compared to athletes from self-paced sports and non-athletes. Importantly, athletes from self-paced sports did not outperform the non-athletes on executive function. The fact that our results do not seem to be in agreement with those of McNeill and colleagues [14] in preschoolers, but do correspond with the findings of Formenti and colleagues [17], whose sample falls within the same age range as the participants of the current study, indicates that differences in executive function might indeed only emerge during late childhood, adolescence, or even young adulthood. Furthermore, the results from Formenti and colleagues [17] also demonstrated that participants from open-skill sports showed better inhibition accuracy than both closed-skill sports participants and a sedentary control group. Additionally, their closed-skill group, which can be compared to the self-paced sports group in the current study, did not outperform the control group on inhibition performance. Consequently, the current study confirms the findings from the limited previous literature within the same age range, and moreover, extends these findings by demonstrating the superiority of young team sports players on a unitary construct of executive function that reflects the performance of seven tasks that measure the different subcomponents of executive function.

In addition, the results from our population of very young athletes seem to be partly consistent with results found in adults, as team sports players outperformed athletes from self-paced sports as well as the non-athletes on the combination of seven different executive function tasks [18,20]. However, our finding that self-paced athletes did not outperform the non-athletic control group contrasts with adult data, where self-paced athletes do outperform a control group on selected measures of executive functioning [21]. Hence, it seems that further (longitudinal) research across the entire lifespan is needed to clarify whether differences found in childhood persist during adolescence and into adulthood. Based on the fact that consistent differences in adults are found between team sports players, self-paced athletes, and non-athletes, it seems plausible that differences that emerge during childhood persist into adulthood and that additional differences (such as superior inhibition for self-paced athletes compared to non-athletes) might emerge later during the further development of executive function, for example in adolescence.

Importantly, the fact that our sample of young athletes from self-paced sports do not outperform a non-athletic control group on executive functioning does seem to support the notion that exercise or physical activity needs to be cognitively challenging to be strongly associated with or provide benefit toward executive function in childhood [13,17]. One could argue that in both self-paced and team sports, young athletes will be cognitively challenged by the need to learn new and complex movement patterns that are inherent to all sports. However, it seems that the highly time-constrained dynamic environment offered by team sports provides that extra layer of cognitive challenge that might be needed to truly be beneficial toward executive function [6]. This could possibly be explained by the fact that participants need to process real-time cues with regard to teammate positions and ball trajectory and constantly update this information in working memory. They also need to be able to inhibit planned actions when that might suddenly not be the best course of action (e.g., passing instead of scoring themselves), and they need to possess great cognitive flexibility to constantly adapt to the dynamic environment that is inherent to team sports [13]. Hence, the findings of the current study provide opportunities for exercise researchers to rethink the nature of their interventions consequently.

A major strength of this study was the use of a weighted sum score derived from seven test scores to assess the construct of executive functioning in a holistic manner. The use of such weighted sum based on a benchmark model that has been validated on a large sample allows us to capture the construct of executive function more adequately, even with a smaller sample size, which precluded running the full benchmark model on the current data set (see Appendix B for a detailed explanation). Nevertheless, it remains important to address the fact that this study, with its cross-sectional nature, was not intended to provide strong conclusions about causality. The current results do not answer the question of whether these team sports athletes demonstrate superior executive functioning because of their involvement in team sports or whether their superior executive function enabled their participation in team sports. Evidently, longitudinal research will be needed to further investigate this issue. Another important aspect that remains to be confirmed is whether this superior executive function performance of team sports athletes during childhood persists across adolescence into adulthood. There is no certainty that the level of executive function measured in our participants will correspond with or predict their executive function levels within two or more years since the participants in our sample are in an important developmental period for executive function [15]. The fact that executive function does indeed show rapid development in our sample is confirmed by the fact that age acted as a significant covariate in our analysis, indicating that even within the narrow age range of 8 to 12 years old, age plays a significant role toward executive functioning. It thus seems valuable to investigate whether this advantage during childhood also evolves into an advantage when development has leveled off in adulthood. While this seems plausible given that comparable results have been found in studies with adults, similar studies including other age groups such as adolescents and young adults still have to be conducted to confirm these findings across the entire lifespan. Furthermore, given the significant influence of age toward executive function during childhood, larger sampled and/or longitudinal studies could explore the development of executive function with age and whether this is influenced by different types of sports participation. Lastly, it should be noted that this study only included girls, and although most studies report no differences in executive functioning between boys and girls at this age [33,34], the results of the current study will need to be confirmed in boys as well.

## 5. Conclusions

In summary, the findings of the current study provide a valuable contribution to the understanding of the relation between youth sports participation and executive function. This study is the first to demonstrate that, even at a very young age, team sports players outperform athletes from self-paced sports as well as non-athletes on a multifaceted and comprehensive test battery for executive function. Additionally, athletes from self-paced sports do not show superior executive functioning compared to non-athletes. Consequently, our findings seem to support the hypothesis that cognitively engaging physical activity, such as participation in team sports, might show stronger associations with executive functioning than other types of sports and physical activity that require less cognitive engagement.

## Figures and Tables

**Figure 1 children-08-00264-f001:**
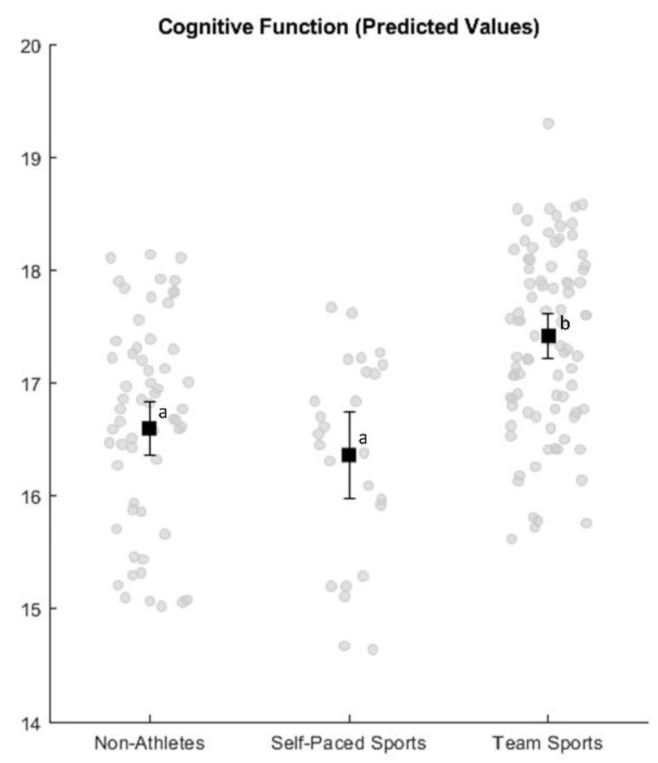
Predicted values (i.e., predicted scores when the influence of the age covariate is taken away) and standard error for executive function within each group. Black squares represent predicted group means, with the bars representing their respective standard errors and the dots representing individual predicted scores. Means with different superscripts are significantly different at the *p* < 0.05 level.

**Table 1 children-08-00264-t001:** Mean age (SD) in years and number of participants in each group.

	Controls	Self-Paced Sports	Team Sports	Total
Age (SD)	10.4 (1.1)	10.3 (1.1)	10.2 (1.0)	10.2 (1.0)
N	59	25	86	170

SD = Standard deviation, N = Number of participants.

**Table 2 children-08-00264-t002:** Mean scores (SD) for each of the tests as well as for the weighted sum for executive function.

	Controls	Self-Paced Sports	Team Sports
Monkey Ladder (MX)	6.27 (1.0)	6.44 (0.9)	6.52 (1.0)
Spatial Span (MX)	4.93 (1.0)	4.62 (0.9)	4.90 (0.9)
Token Search (MX)	6.37 (1.4)	6.09 (1.6)	6.73 (1.5)
Double Trouble (%)	63.38 (12.9)	62.39 (12.2)	61.66 (13.7)
Sustained Attention to Response (%)	35.03 (20.8)	41.28 (16.0)	39.21 (20.4)
Odd One Out (CA)	14.41 (1.9)	14.16 (2.4)	15.21 (2.3)
Spatial Planning (SC)	16.85 (8.0)	16.76 (4.5)	16.12 (5.7)
Cognitive Functioning Weighted Sum Score	16.59 (2.0)	16.36 (2.1)	17.42 (2.0)

MX = maximum recall, % = percent correct responses, CA = correct attempts, SC = Score.

## Data Availability

Data from this study can be obtained by request to the authors.

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
