# Peer review of "Children Involved in Team Sports Show Superior Executive Function Compared to Their Peers Involved in Self-Paced Sports"

_children, 2021, doi:10.3390/children8040264_

Round 1
Reviewer 1 Report
I thought the paper was well-written and organized overall. I Found the introduction clear and concise, with some minor improvements to wording/structure needed in the first few paragraphs. The authors are clear in describing the strengths and limitations of the study. It does appear that the overall effect size is relatively small, and it would make sense to include a sentence or two on the real-world implications of the study in that context.
Author Response
Please see the attachment.
Comments and Suggestions for Authors
I thought the paper was well-written and organized overall. I Found the introduction clear and concise, with some minor improvements to wording/structure needed in the first few paragraphs. The authors are clear in describing the strengths and limitations of the study. It does appear that the overall effect size is relatively small, and it would make sense to include a sentence or two on the real-world implications of the study in that context.
We thank the reviewer for these comments. The effect size in this study can be considered as borderline moderate rather than small according to the Cohen (1988) cut-offs. This has now been added to the manuscript (lines 191-193). In addition, we feel that it is difficult to put forward real-world implications without investigations of causality, regardless of our effect size. The most important implication might be the notion that the type of organized sports a child participates in is associated with executive function, and further investigation is thus necessary to deepen our understanding of this association at different ages, which is discussed at large in the discussion in the following sections:
Line 230 […] “Hence, it seems that further (longitudinal) research across the entire lifespan is needed to clarify whether differences found in childhood persist during adolescence and into adulthood. Based on the fact that consistent differences in adults are found between team sports players, self-paced athletes and non-athletes, it seems plausible that differences that emerge during childhood persist into adulthood, and that additional differences (such as superior inhibition for self-paced athletes compared to non-athletes) might emerge later during the further development of executive function, for example in adolescence. “
Line 252 […] “Hence, the findings of the current study provide opportunities for exercise researchers to rethink the nature of their interventions consequently.”
Line 261 […] “The current results do not answer the question of whether these team sports athletes demonstrate superior executive functioning because of their involvement in team sports, or whether their superior executive function enabled their participation in team sports. Evidently, longitudinal research will be needed to further investigate this issue. Another important aspect that remains to be confirmed is whether this superior executive function performance of team sports athletes during childhood persists across adolescence into adulthood. There is no certainty that the level of executive function measured in our par-ticipants will correspond with or predict their executive function levels within two or more years, since the participants in our sample are in an important developmental peri-od for executive function”

Reviewer 2 Report
1.- The introduction should delve into the brain areas that stimulate team sports activity and the brain areas that we activate when we perform a task that requires executive functions. Much has been written about the latter and hardly any works are cited.
2.- It is also necessary to expand the exposition on what executive functions are, what kind of tasks are involved and how they are measured, what tools are commonly used.
3.- The strategy followed to collect the data must be described more fully.
4.- Describe better what type of sports are included in the two groups of girls who carry out frequent sports practice.
5.- Indicate the type of study design.
6.- Present the psychometric properties of the tools they use. Better describe the tools.
7.- Clarify the wording of the data analysis section, with the analyzes that have been used and duly referenced. In addition, since they have verified the assumption of homoscedasticity (necessary for ANCOVA; although for this test, Levene is not the best test to assess the homogeneity of variances, but rather the highest and lowest standard deviations should be fixed and indicated in the paragraph with respect to the variable EF, which seems to have been divided into many dimensions).
8.- It would be interesting for them to also check the assumptions of normality and independence to see what sampling distribution the data has and also check its randomness (in this way, the data are much more reinforced at a methodological and clinical level). The most important thing in this section is that they clarify the analyzes well, what they use them for and that they support them with references.
9.- No statistical test is referenced and neither do they comment on the cut-off points for the size of the effect, or who took them.
10.- On the other hand, in the objective they put "predict". It is clear that ANCOVA is a mix between ANOVA and a regression, but I don't know if the best way to bring the article together would be to do some moderation analysis. With ANCOVA they control for confounding variables, not bad, but the terms used in the objective make me a bit dizzy.
Cheer up¡
Reviewer 3 Report
- This study compared executive function performance of children aged 8 to 12 that either participated in team sports or self-paced sports, or were not involved in any kind of organized sports (non-athletes). The study is very interesting and I congratulate the authors in this regard, however, it is quite clear and self-evident that the children who participate in team sports / self-paced sports, should have a higher executive function compared to non-athletes.
- I suggest that the authors in the Introduction chapter to clearly delimit what athletes who are engaged in team sports mean; athletes from self-paced sports; and representative peers who are not involved in sports. It is very good to understand from the beginning what these things mean, many readers may not know exactly what they mean.
- Which was the reasoning for which only female subjects were enrolled in this study, please detail.
- The fact that the girls who participated in ''Self-Paced Sports: for at least 2 hours per week'' is quite small compared to the other two categories, i.e. non-Athletes and Team Sports, do not statistically affect research results? I think the difference in the number of girls is quite large when we talk about self-paced sport, i.e 25......and for the other two categories the number is 59 and 86.
- In the results chapter I would introduce other data regarding the research, I think that there is too little information provided in this chapter, I see Mean scores (SD) in table 1, Levene’s test, Bonferroni etc. Please include and other data.
Round 2
Reviewer 2 Report
The work is correct. I hope you can continue collecting samples and include the male gender in your future studies, surely the data can be very interesting when comparing genders.
Author Response
Thank you for this comment, we certainly aim to continue this work and include male participants in the future.
Reviewer 3 Report
Comments:
-
It is interesting to note that no EF differences were found between children participating in self-paced sports and non-athletes.
Specifically, the authors can detail why there is no such difference, I basically think there should be differences, it is strange for me to believe that the differences are similar.Also, I do not think that one study is enough to support this statement, I think it is too little.If there are other studies that confirm this, we can say that the authors' statements are relevant. - Otherwise, the authors made the required changes
Author Response
Indeed, based on the results in adults (e.g. Jacobson & Matthaeus, 2014), we would have expected to see those participating in self-paced sports outperforming non-athletes. However, only one other study has investigated this issue in children before us (Formenti et al., 2021), and they also did not find differences between self-paced athletes and non-athletes between 8 and 12 years old. The fact that there is only one other study investigating this in children strongly highlights the value of this work and the need for further research in this domain.
We elaborate on this issue in the discussion (lines 245 - 271), where we state two possible reasons for the lack of difference between self-paced athletes and non-athletes. In short, the fist reason might be that these differences between self-paced sports and non-athletes emerge only later in life. Secondly, in line with suggestions of Pesce and colleagues, the more dynamic and complex environment of team sports might provide more opportunities to develop superior executive function compared to self-paced sports.
Line 245: "However, our finding that self-paced athletes did not outperform the non-athletic control group contrasts with adult data, where self-paced athletes do outperform a control group on selected measures of executive functioning [21]. Hence, it seems that further (longitudinal) research across the entire lifespan is needed to clarify whether differences found in childhood persist during adolescence and into adulthood. Based on the fact that consistent differences in adults are found between team sports players, self-paced athletes and non-athletes, it seems plausible that differences that emerge during childhood persist into adulthood, and that additional differences (such as superior inhibition for self-paced athletes compared to non-athletes) might emerge later during the further development of executive function, for example in adolescence.
Importantly, the fact that our sample of young athletes from self-paced sports do not outperform a non-athletic control group on executive functioning does seem to support the notion that exercise or physical activity needs to be cognitively challenging to be strongly associated with or provide benefit towards executive function in childhood [13,17]. One could argue that in both self-paced and team sports, young athletes will be cognitively challenged by the need to learn new and complex movement patterns that are inherent to all sports. However, it seems that the highly time-constrained dynamic environment offered by team sports provides that extra layer of cognitive challenge that might be needed to truly be beneficial towards executive function [6]. This could possibly be explained by the fact that participants need to process real-time cues with regard to teammate positions and ball trajectory, and constantly update this information in working memory. They also need to be able to inhibit planned actions when that might suddenly not be the best course of action (e.g. passing instead of scoring themselves), and they need to possess great cognitive flexibility to constantly adapt to the dynamic environment that is inherent to team sports [13]. Hence, the findings of the current study provide opportunities for exercise researchers to rethink the nature of their interventions consequently."